# Can FDG-PET/CT imaging be used to predict decline in quality of life in interstitial lung disease? A prospective study of the relationship between FDG uptake and quality of life in a UK outpatient setting

Louise Helen Jordon [1,2] Balaji Ganeshan,[3] Iftikhar Nadeem,[2] Luke Hoy,[3] Noor Mahdi,[2] Joanna C Porter,[4,5] Ashley Groves,[3] Thida Win[6]

This work was presented at the 2022 European Respiratory Society International Congress in the ILD poster session. Eur Respir J 2022; 60: Suppl.66, 2473. https://doi.org/10.1183/13993003.congress-2022.2473.

For numbered affiliations see end of article.

**Correspondence to**
Dr Louise Helen Jordon;
louise.jordon@nhs.net

## ABSTRACT

**Background** [18]Fluorine-fluorodeoxyglucose ([18]F-FDG) positron emission tomography (PET) CT imaging has been used in many inflammatory and infectious conditions to differentiate areas of increased metabolic activity. FDG uptake differs between areas of normal lung parenchyma and interstitial lung disease (ILD).

**Objectives** In this study, we investigated whether FDG-PET/CT parameters were associated with a change in the quality of life (QoL) in patients with ILD over 4 years of follow-up.

**Methods** Patients underwent PET-CT imaging at diagnosis and were followed up with annual QoL assessment using the St George's Respiratory Questionnaire (SGRQ) until death or 4 years of follow-up. Maximum standard uptake value (SUVmax) and Tissue-to-Background Ratio (TBR) were assessed against SGRQ overall and subscale scores.

**Results** 193 patients (94 patients in the idiopathic pulmonary fibrosis (IPF) subgroup and 99 patients in the non-IPF subgroup) underwent baseline FDG-PET/CT imaging and QoL assessment. Weak-to-moderate correlation was observed between baseline SUVmax and SGRQ scores in both ILD subgroups. No relationship was observed between baseline SUVmax or TBR and change in SGRQ scores over 4 years of follow-up. In the IPF subgroup, surviving patients reported a decline in QoL at 4 years post diagnosis whereas an improvement in QoL was seen in surviving patients with non-IPF ILD.

**Conclusions** Weak-to-moderate positive correlation between baseline SUVmax and SGRQ scores was observed in both ILD subgroups (IPF:$r_s$=0.187, p=0.047, non-IPF: $r_s$=0.320, p=0.001). However, baseline SUVmax and TBR were not associated with change in QoL in patients with IPF and non-IPF ILD over 4 years of follow-up. At 4 years post diagnosis, surviving patients with IPF reported declining QoL whereas improvement was seen in patients with ILD who did not have IPF.

---

### STRENGTHS AND LIMITATIONS OF THIS STUDY

⇒ Large sample of patients with idiopathic pulmonary fibrosis (IPF) and non-IPF interstitial lung disease (ILD).

⇒ Four years of quality of life follow-up data for patients with good questionnaire return rates in surviving patients.

⇒ Effect of treatment and specific ILD subtype (aside from IPF) data are not assessed.

---

characterised by varying degrees of fibrosis and inflammation of the lung parenchyma. ILDs are most frequently diagnosed by a combined clinical and radiological assessment.[1] Of these, idiopathic pulmonary fibrosis (IPF) and non-specific interstitial pneumonia (NSIP) are the most frequently reported, with progressive fibrosis as the dominant histopathological lesion in IPF, and areas of inflammation and, in some cases, fibrosis occurring in NSIP.[2]

Despite antifibrotic treatments, the prognosis for IPF is poor, with a mean survival of 2.5–5 years from diagnosis.[3] Quality of life (QoL) in IPF is affected by severe breathlessness and activity impairment,[4] which demonstrates a close relationship with changes in lung function.[5] Prognosis and symptom burden in NSIP are much better, with an 86%–98% 5-year survival for certain subtypes.[6]

At diagnosis and during follow-up, it is difficult to predict prognosis for patients with ILD. When asked, patients have reported outcomes which they value most to be symptom burden, daily functioning and QoL.[7] Presently, decline in forced vital capacity (FVC) is most frequently used to monitor disease

## INTRODUCTION

Interstitial lung diseases (ILDs) are a heterogeneous group of pulmonary conditions

progression,[8] however, baseline FVC is not predictive of change in QoL.[5] Routinely used biomarkers are lacking,[9] and so tools to accurately predict QoL deterioration are needed.

Positron emission tomography (PET) combined with CT offers a minimally invasive investigation to locate and quantify cellular metabolism. The most frequently used PET tracer is the glucose analogue, fluorodeoxyglucose (FDG). When labelled with the radioisotope 18-Fluorine ([18]F), it can be used to visualise areas of regional glucose uptake and indicate metabolic activity.[10–12]

Tissue metabolic activity on PET-CT is quantified through standardised uptake values (SUVs) which measure the tracer's radioactivity concentration relative to the dose given and the patient's weight.[13] The SUV is most commonly reported as the maximum SUV within the region of interest (SUVmax).[14] The tissue-to-background ratio (TBR) is a measure of variation of FDG uptake within the lung, where a value close to one indicates relatively uniform uptake.[15]

In IPF lung tissue, increased glycolysis has been observed in fibroblasts,[16] with increased FDG uptake seen in areas of honeycombing.[17 18] We have previously shown that FDG uptake on PET/CT is a prognostic biomarker in IPF[19] and showed TBR to be predictive of 3 year mortality. In patients with fibrotic ILD, we have observed that TBR correlates with neoangiogenesis markers,[20] and in patients with NSIP we observed that a high pulmonary background signal (SUVmin) negatively correlated with mortality.[21] Others have investigated the utility of FDG uptake in ILD subtype differentiation[22] and in determining disease activity in ILDs associated with rheumatological conditions.[23]

The primary aim of this study was to investigate any relationship between baseline FDG-PET signal (measured by SUVmax, SUVmin and TBR) and QoL (as a self-reported questionnaire) at the time of diagnosis and longitudinally over a 4-year follow-up period in patients with ILD, to determine the utility of [18]F-FDG PET/CT in ILD QoL prognostication.

## MATERIALS AND METHODS
### Study setting and patient selection
This was a prospective, single-centre study from one UK hospital. Patients were consecutively recruited from a general respiratory clinic and enrolled for 4 years during the study period from November 2009 to December 2020. All patients with ILD who consented to having a PET-CT scan and being followed up for 4 years were eligible for inclusion in the study.

All patients underwent full clinical assessment including multidisciplinary team (MDT) review, baseline pulmonary function tests and high-resolution CT (HRCT) evaluation. Infection and neoplasia were excluded on clinical and radiological grounds. The diagnosis of IPF was made by MDT review and was based on HRCT findings of a 'usual interstitial pneumonia' pattern with no associated

cause. If patients did not meet these diagnostic criteria, they were grouped as non-IPF ILD. Written informed consent was obtained for all participants.

Patients had an [18]F-FDG PET/CT scan and completed a concurrent baseline QoL questionnaire. They were subsequently followed for 48 months with QoL questionnaires at 12, 24 and 48 month intervals. Patients were excluded if they did not complete a valid QoL questionnaire at baseline, if the SUVmax and TBR could not be calculated or if a diagnosis of IPF or non-IPF ILD could not be reached. A valid questionnaire was one completed in accordance with instructions, with enough questions answered to generate a score.

Baseline demographic and pulmonary function test (PFT) data were also collected. Repeated pulmonary function tests were attempted at 12, 24 and 48 months of follow-up. Baseline ILD treatment data were available for a subset of patients. Baseline treatment was regarded as treatment prescribed within the 12 months up to and including the study commencement date.

In total, 234 patients with ILD were recruited and had [18]F-FDG PET/CT scans with quantification of SUVmax, SUVmin and TBR. 41 patients were excluded as they did not complete a valid baseline questionnaire, leaving 193 patients in the study.

### Study objective
The primary aim of this study was to investigate the relationship between baseline [18]F-FDG PET/CT pulmonary uptake (SUVmax and TBR) and QoL measures, assessed using the St George's Respiratory Questionnaire (SGRQ),[24] at the time of the scan and at 12, 24 and 48 months postbaseline.

Secondary objectives included looking for correlation between FDG uptake and QoL subscale scores in the total study population and in IPF and non-IPF groups, and observing QoL trajectory over the study period.

### QoL data collection
The SGRQ is a disease-specific questionnaire designed to measure the impact on overall health, daily life and perceived well-being in patients with respiratory disease.[24 25] It has 50 questions divided into three sections covering: respiratory symptoms, activities that cause or are limited by breathlessness and the psychosocial impact of disease. Each question has an empirically derived weight,[24] and results in a subsection and overall score out of 100, with higher scores reflecting increased severity. The SGRQ has been well validated in UK and international populations.[24 25]

Baseline SGRQ data were collected when patients attended for their [18]F-FDG PET/CT scan (T0), and then at 12, 24 and 48 months later (T12, T24 and T48), by postal questionnaire. QoL trends were calculated by subtracting the baseline QoL score from the score at a subsequent timepoint, thus a positive or negative score indicated a deterioration or improvement, respectively.

## FDG PET-CT imaging data collection

The $^{18}$F-FDG PET/CT was performed as previously described.[15]

Patients fasted for 6 hours. Images were acquired 1 hour after an injection of $^{18}$F-FDG (200MBq) on a dedicated combined PET/64-detector CT scanner (GE Healthcare). Three sequential scans of the thorax were performed while the patient remained supine on the table with their arms above their head. First, an attenuation-correction CT (CTAC) was performed using 64 3.75 mm detectors, a pitch of 1.5, and a 5 mm collimation (140kVp and 80mA in 0.8 s).

Patient position was maintained while a whole-body 18F-FDG PET scan was acquired, covering an area identical to the CTAC. All images were acquired in two-dimensional mode (8 min/bed position). Transaxial emission images of 3.27 mm thickness (pixel size 3.9 mm) were reconstructed using ordered-subsets expectation maximisation with 2 iterations and 28 subsets. The axial field of view was 148.75 mm, resulting in 47 slices per bed position. Next, a deep inspiratory HRCT was performed while maintaining patient position, using 64 1.25 mm detectors, pitch of 0.53 and 1.25 mm collimation (120 kVp and 100 mAs).

## Image analysis

PET/CT images were reviewed by two combined radiologist/nuclear medicine physicians in consensus, one of whom had a specialist interest in cardiothoracic PET/CT. PET/CT images were loaded onto an Xeleris workstation (GE Healthcare Technology). Both attenuation and non-attenuation-corrected images were reviewed to ensure that areas of higher-density lung were not inducing attenuation-correction artefacts. The area of most intense pulmonary FDG uptake was identified and SUVmax measured. The HRCT parenchymal pattern in the region was also assessed. In addition, SUVmin was used to determine the background lung uptake and in turn, calculate TBR.

| Table 1 | Baseline characteristics of the study population |
|---|---|
| | **Data** |
| Males (%) | 131 (67.8) |
| Mean age (SD) | 66.6 (9) |
| Diagnosis: IPF (%) | 94 (48.7) |
| NSIP (%) | 78 (40.4) |
| Non-NSIP (%) | 21 (10.9) |
| Mean baseline FEV1 (SD; range) | 74.9 (18.7; 17–122) |
| Mean baseline FVC (SD; range) | 75.8 (17.6; 20–129) |
| Mean baseline TLCO (SD; range) | 46.7 (15.5; 11–98) |

FEV1, forced expiratory volume in 1 s; FVC, forced vital capacity; IPF, idiopathic pulmonary fibrosis; NSIP, non-specific interstitial pneumonia; TLCO, Transfer Factor.

## Statistical analysis

Statistical analyses were performed using SPSS V.26 (IBM) and Prism V.8 (GraphPad). A two-sided p<0.05 indicated a significant relationship. The relationship between FDG-PET/CT uptake and QoL was assessed using non-parametric Spearman's rank correlation.

## Patient and public involvement

While patients were not involved directly in the design of this study, the research question and outcome measures were focused on QoL rather than lung function or other physical parameters in line with patients' greatest priorities.

## RESULTS

### Total population

#### Baseline characteristics

In total, 193 patients met the inclusion criteria (131 males, 62 females); average age±SD: 66.6±9 years. Table 1 shows baseline characteristics.

The mean values for the baseline PET uptake (range in brackets) for the population were: SUVmax 3.25 (1.4–7.4), SUVmin 0.55 (0.16–1.3) and TBR 6.28 (2.4–15.3).

During follow-up of the 193 patients, 21 died by 12 months (10.9%), 42 by 24 months (21.8%) and 120 by 48 months (62.2%).

#### QoL assessment

QoL data were available for all 193 patients at the time of recruitment (T0), 99 patients at T12 (57.2% surviving patients), 91 patients at T24 (60.3% surviving patients) and 45 patients at T48 (61.6% surviving patients).

Online supplemental table S1 demonstrates the mean QoL scores for the population who returned the SGRQ at the study time points, displayed as a percentage of the total possible score.

There was no significant difference in total SGRQ scores at T0 and T12 (t=0.44, p=0.66), T24 (t=0.36, p=0.72) or T48 (t=0.2, p=0.84).

#### Relationship between FDG-PET and SGRQ

Example PET-CT images with their corresponding SGRQ scores are shown in figure 1A–D.

At baseline, there was weak positive correlation between SUVmax and total SGRQ score ($r_s$=0.229, p=0.002) and the SGRQ Symptom, Activity and Impact subscales ($r_s$=0.211, 0.178 and 0.209, respectively, p<0.003 for all). There was also weak positive correlation between SUVmin and baseline total SGRQ score ($r_s$=0.233, p=0.001) and the SGRQ Symptom, Activity and Impact subscales ($r_s$=0.161, 0.153 and 0.236, respectively, p<0.05 for all). There was no significant relationship between TBR and total ($r_s$=−0.015, p=0.841) and subscale SGRQ scores.

When a change in QoL over time was assessed, there was no significant correlation between change in total SGRQ score and baseline SUVmax at 12 months ($r_s$=−0.084, p=0.403), 24 months ($r_s$=0.184, p=0.084) and 48 months ($r_s$=0.084, p=0.56). Additionally, there was no relationship

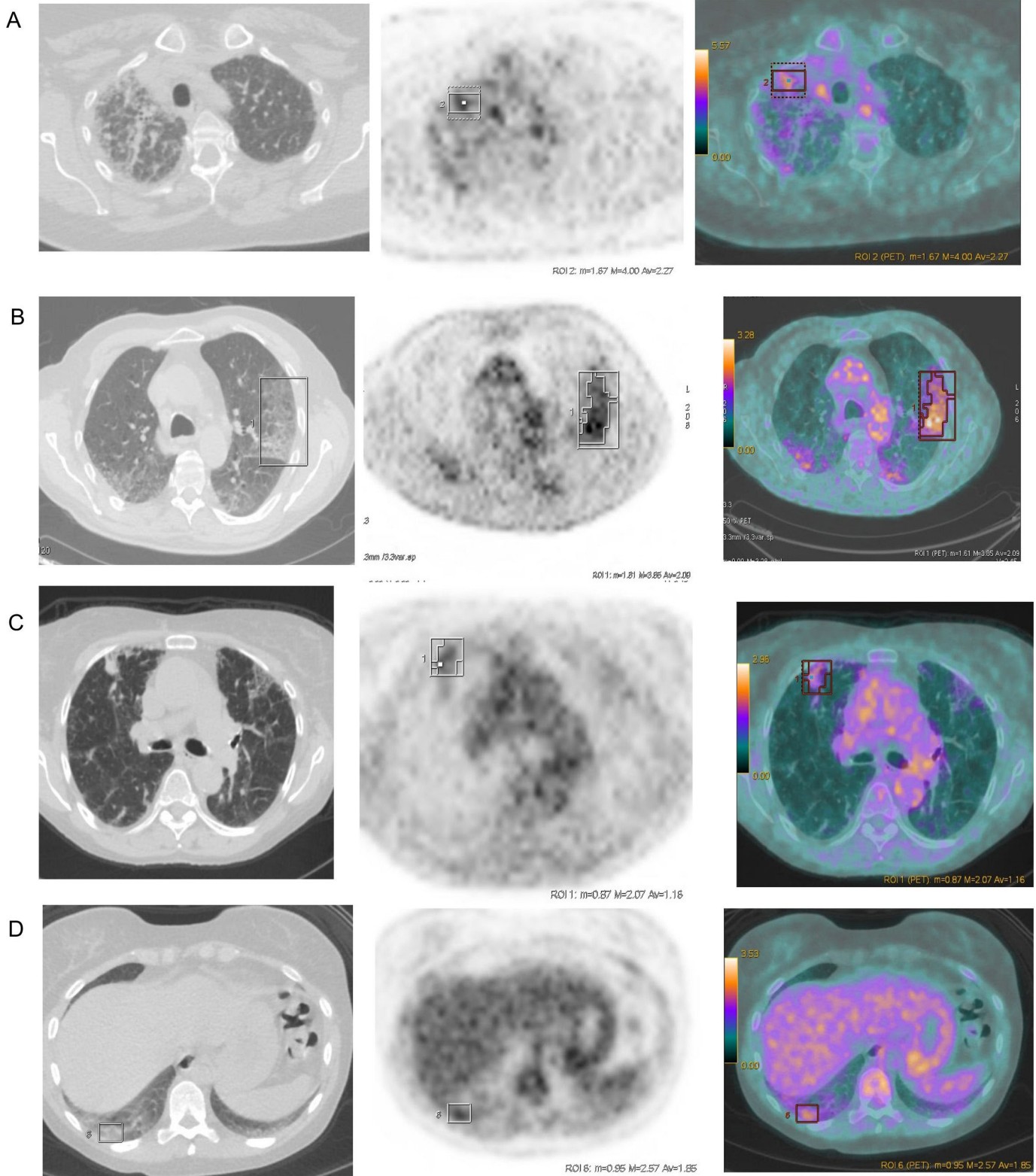

**Figure 1** Example PET-CT images from study patients. Images show the helical acquisition CT scan, attenuation-corrected F18 emission and the axially fused PET-CT (L→R). Boxed area on the scans indicates region of SUVmax. (A) SUVmax 4.1, SGRQ 72.99, (B) SUVmax 3.9, SGRQ 58.83, (C) SUVmax 2.1, SGRQ 7.85, (D) SUVmax 2.68, SGRQ 15.66. PET, positron emission tomography; SGRQ, St George's Respiratory Questionnaire; SUVmax, maximum standard uptake value.

between change in SGRQ subscale scores at each time point and SUVmax. There was no significant correlation between change in total SGRQ score at 12 months ($r_s$=−0.017, p=0.863), 24 months ($r_s$=0.122, p=0.253 and 48 months ($r_s$=0.140, p=0.332) and baseline SUVmin. We observed no significant correlation between change

in total SGRQ score and baseline TBR at T12 ($r_s$=−0.065, p=0.519), T24 ($r_s$=−0.014, p=0.893) and T48 ($r_s$=−0.113, p=0.435), and no relationship between change in SGRQ subscale scores at each time point and TBR.

### Relationship between FDG-PET, SGRQ and other parameters

Age at enrolment did not correlate with baseline SUVmax ($r_s$=−0.103, p=0.158) or baseline TBR ($r_s$=−0.033, p=0.646).

Baseline FVC and Transfer Factor (TLCO) data were available for 185 and 163 patients, respectively. Baseline PET parameters did not correlate with baseline PFT data, nor did baseline PET parameters correlate with change in PFT data after 12, 24 or 48 months of follow-up. Further details are included in the supplemental materials (online supplemental table S2).

ILD treatment data were available for 127 participants at baseline (127/193; 66%). Of these, 84 participants were on no treatment, 35 were on steroid treatment, 9 were on antifibrotic treatments and 10 were on immunosuppressant treatment. 13 patients were on more than one treatment at baseline, 11 of which comprised treatments in different categories. In this subgroup of patients where treatment data were available, type of treatment (or no treatment) at baseline did not affect baseline SUVmax (H(4)=0.88, p=0.83) or SUVmin (H(4)=4.41, p=0.22). There was a significant difference in median TBR between the treatment groups (median TBR by group: no treatment=6.0, steroid=5.0, antifibrotic=5.3, immunosuppressant=4.0, H(4)=8.54, p=0.036.

There was weak negative correlation between baseline FVC and baseline TLCO and total baseline SGRQ score (FVC: $r_s$=−0.285, p<0.001, TLCO: $r_s$=−0.214,p=0.006). Change in PFT measures did not correlate with change in total SGRQ score over the 48 month follow-up period.

### IPF- ILD subset
### Baseline demographics

There were 94 patients (84 males); average-age±SD: 69.5±8.9 years) in the IPF-ILD subgroup.

The mean (range) of baseline PET metrics was SUVmax 3.4 (1.8–6.0) and TBR 5.4 (3.1–16.6).

### QoL assessment

QoL data were available in all 94 patients at T0. At 12 months, 47 patients (50%) returned the SGRQ, 40 (42.5%) at 24 months and 22 (23.4%) at 48 months. During the follow-up period for the 94 patients, 13 died by 12 months (13.8%), 24 by 24 months (25.5%) and 74 by 48 months (78.7%).

Online supplemental table S3 shows the total and subscale SGRQ scores at each time point for the IPF subgroup.

When a change in QoL was assessed over time, there was no significant difference in total SGRQ at T0 and T12 (t=0.74, p=0.46) or T0 and T24 (t=1.23, p=0.22) but a significant increase in total SGRQ, indicating deteriorating QoL, was seen at 48 months (t=−3.1. p=0.002).

### Relationship between SGRQ scores and FDG-PET

At baseline, there was weak positive correlation between total SGRQ score and SUVmax ($r_s$=0.187, p=0.047; figure 2). There was also weak positive correlation between the Activity SGRQ subscale and SUVmax ($r_s$=0191, p=0.039). There was no significant relationship between TBR and total ($r_s$=0.061, p=0.524) and subscale SGRQ scores, and no significant relationship between SUVmin and total ($r_s$=0.107, p=0.26) and subscale SGRQ scores.

When a change in QoL over time was assessed against pulmonary FDG-PET uptake, there was no significant correlation between change in total SGRQ score and baseline SUVmax at T12 ($r_s$=0.065, p=0.653), T24 ($r_s$=0.017, p=0.913) and T48 ($r_s$=0.084, p=0.684; figure 1). Additionally, there was no relationship between change in SGRQ subscale scores at each time point and SUVmax, and no relationship between total or subscale SGRQ scores and SUVmin.

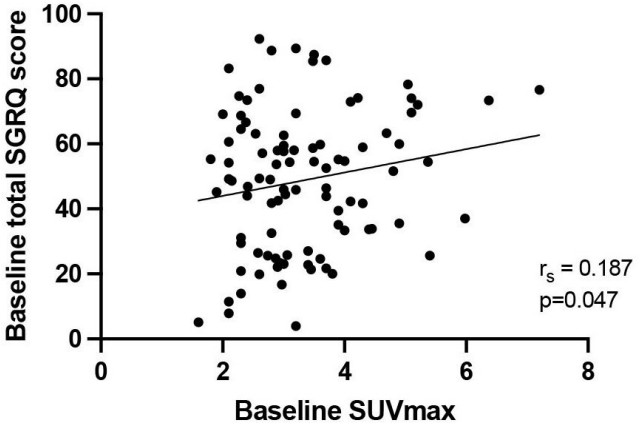

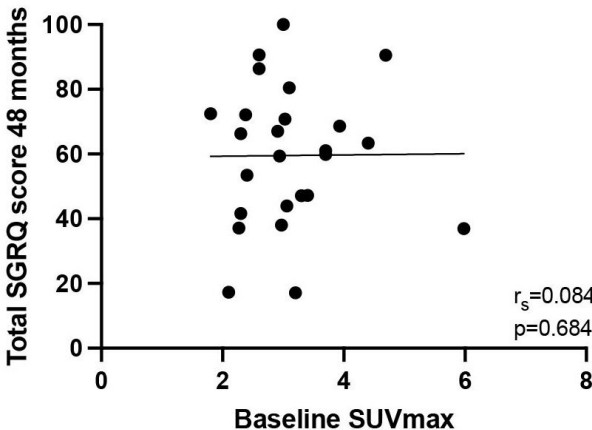

**Figure 2** Relationship between baseline SUVmax and SGRQ at (A) baseline and (B) 48 months follow-up in the IPF subgroup. IPF, idiopathic pulmonary fibrosis; SGRQ, St George's Respiratory Questionnaire; SUVmax, maximum standard uptake value.

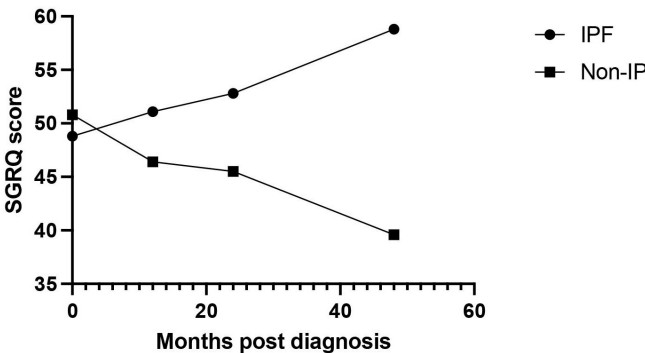

**Figure 3** SGRQ by ILD group over 48-month follow-up period. ILD, interstitial lung disease; IPF, idiopathic pulmonary fibrosis; SGRQ, St George's Respiratory Questionnaire.

We also observed no significant correlation between change in total SGRQ score and baseline TBR at T12 ($r_s$=0.035, p=0.807), T24 ($r_s$=−0.122, p=0.419) and T48 ($r_s$=0.006, p=0.978, and no relationship between change in SGRQ subscale scores at each time point and TBR.

### Non-IPF subset
#### Baseline demographics
There were 99 patients (45 males) in the non-IPF ILD subgroup; mean age±SD: 64.6±10.1 years. The mean (range) of baseline PET uptake metrics was SUVmax 3.1 (1.4–6.4) and TBR 6.8 (2.9–13.6).

#### QoL assessment
QoL data were available in all 99 patients at the time of recruitment (T0). 52 patients (52.5%) returned the QoL questionnaire at 12 months, 51 (54.3%) at 24 months and 23 (23.2%) at 48 months. Of the 99 patients, 8 died by 12 months (8.1%), 18 by 24 months (18.1%) and 46 by 48 months (46.5%) of follow-up.

Online supplemental table S4 shows the total and subscale SGRQ scores at each time point for the non-IPF subgroup.

Whe change in QoL was assessed over time, there was no significant difference in total SGRQ at T0 and T12 (t=1.14, p=0.26) or T0 and T24 (t=1.23, p=0.22) but a significant decrease in total SGRQ, indicating improved QOL, was seen at 48 months (t=2.04. p=0.05).

In comparison with total SGRQ scores from the IPF subgroup, there was no significant difference at T0 (t=0.614, p=0.54), T12 (t=1.09, p=0.28) or T24 (t=1.43, p=0.16). Scores in the non-IPF group were significantly lower than those of the IPF group at T48 (t=−3.15, p=0.003). Comparison of overall SGRQ scores between the IPF and non-IPF subgroups is shown in figure 3.

#### Relationship between SGRQ scores and FDG-PET
At baseline, there was a moderate positive correlation with the total SGRQ score and SUVmax ($r_s$=0.320, p=0.001; figure 4). There was also positive correlation between the baseline Symptoms, Activity and Impact subscale scores and baseline SUVmax ($r_s$=0.287, 0.242 and 0.314, p<0.008, respectively). As baseline, there was a moderate positive correlation between total SGRQ and SUVmin ($r_s$=0.390, p=0.001). There was also positive correlation between the baseline Symptoms, Activity and Impact subscale scores and baseline SUVmin ($r_s$=0.230, 0.212 and 0.366, p<0.01, respectively). There was no significant relationship between TBR and total ($r_s$=−0.085, p=0.379) and subscale SGRQ scores.

When a change in QoL over time was assessed against pulmonary FDG PET uptake, there was no significant correlation between change in total SGRQ score and baseline SUVmax at T12 ($r_s$=−0.126, p=0.347), T24 ($r_s$=0.188, p=0.188) and T48 ($r_s$=−0.121, p=0.539; figure 3). Additionally, there was no relationship between change in SGRQ subscale scores at each time point and baseline SUVmax, and no relationship between change in total or subscale SGRQ scores and SUVmin over time.

We also observed no significant correlation between change in total SGRQ score and baseline TBR at T12

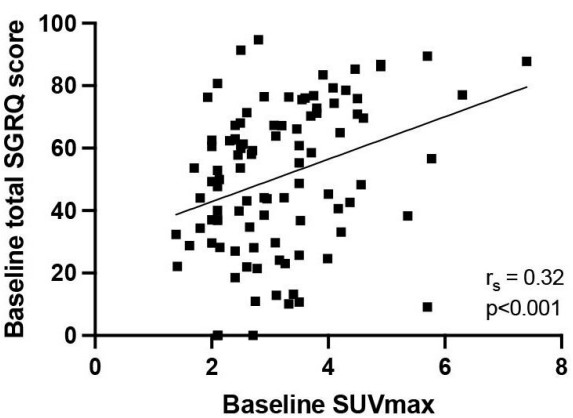

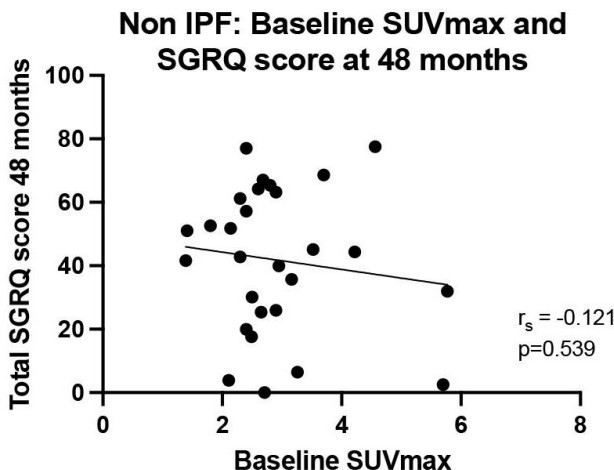

**Figure 4** Relationship between baseline SUVmax and SGRQ at (A) baseline and (B) 48 months follow-up in the non-IPF subgroup. IPF, idiopathic pulmonary fibrosis; SGRQ, St George's Respiratory Questionnaire; SUVmax, maximum standard uptake value.

($r_s$=−0.051, p=0.702), T24 ($r_s$=0.075, p=0.603) and T48 ($r_s$=0.083, p=0.676, and no relationship between change in SGRQ subscale scores at each time point and TBR.

## DISCUSSION

The aim of this study was to investigate the relationship between [18]FDG-PET/CT metrics and QoL in patients with IPF and non-IPF ILD to determine if [18]FDG-PET/CT could be used in ILD QoL prognostication. To our knowledge, this study is the first and largest to date investigating this relationship.

### Relationship between FDG-PET/CT and QoL

Our study showed that there was a weak association between baseline SUVmax and QoL measured by the SGRQ. This may indicate that the metabolic activity of an ILD may relate to a patient's symptoms and QoL. However, in both the IPF and non-IPF subgroups, we observed no significant relationship between baseline SUVmax or TBR and change in QoL over the 48-month follow-up period. As such, these baseline PET measurements do not appear to be associated with a patient-reported change in QoL for either ILD subgroup.

Work from our group has previously shown higher TBR to be predictive of survival in patients with IPF.[19] More recently, we have also shown TBR to correlate with biomarkers of new vessel angiogenesis, which itself is predictive of poorer survival.[20] In 27 patients with IPF, SUVmax was shown to be predictive of FVC, TLCO and physiological decline at 12 months.[18] A larger study of 89 patients reported baseline SUVmax to be predictive of TLCO decline at 6 months, but no relationship was observed with survival outcomes.[26]

However, existing literature on FDG-PET/CT as a prognostic tool in IPF is mixed. Groves et al[17] observed a relationship between baseline SUVmax and baseline global health and lung function in 18 patients with IPF, but these measures were not predictive of decline at 6 months. This concurs with our findings of a correlation between baseline SUVmax and QoL measures in IPF patients. When evaluating the role of [18]F-FDG PET/CT in predicting treatment response in IPF, Bondue et al found SUVmax to be predictive of pirfenidone response in a mouse bleomycin model, but not in 25 patients with IPF, and nor was it predictive of disease progression during the 3-month study period in the patient population.[27]

Small previous studies have shown FDG uptake distribution in NSIP, but not SUVmax to be predictive of response to treatment.[28] In patients with advanced systemic sclerosis associated ILD, [18]F-FDG PET/CT was unable to differentiate between areas of active inflammation (ground glass opacities on HRCT) and areas of fibrosis but could differentiate between normal lung parenchyma and ILD changes.

To our knowledge, this is the first study evaluating the relationship between patient-reported QoL and SUVmax and TBR. While some studies have shown a relationship

between these parameters and lung function or survival, we may not have seen a relationship between them and QoL trajectory because QoL has many more contributing factors. Furthermore, the drop-out in questionnaire returns over our study period is due in part to patient death, and hence only patients with a milder disease trajectory were evaluated at our 48 month time point. Additionally, while baseline forced expiratory volume in 1 s and FVC measures of our patients were similar to those in other studies,[18] our patients were enrolled from diagnosis whereas other studies enrolled subjects at later disease stages. Finally, studies evaluating PET-CT as a prognostic tool have focused on different PET parameters such as FDG uptake pattern and SUVmean, and hence using these may have led to an observed relationship with QoL.

Combining this study with prior work,[5 17 19] there is presently insufficient evidence to support using baseline FVC[5] or SUVmax/TBR to predict symptomatic decline in ILD,[17] but these potentially can be used to predict survival outcomes in IPF.[19]

### QoL trajectory in patients with IPF and non-IPF ILD

The trajectory of patients' QoL over the follow-up period was evaluated as one of our secondary outcomes. Better QoL was observed in the non-IPF subgroup compared with the IPF group at all points from 12 to 48 months follow-up. In the IPF subgroup, QoL deterioration was observed at the 48 month time point, but not prior to this. Notably, 78% of the initial IPF cohort had died at 48 months. For the non-IPF group, there was no difference in QoL measures at 12 and 24 months compared with baseline, but a slight improvement seen at 48 months, by which point 46% of the cohort had died. Thus, patients with IPF who survived to 48 months postdiagnosis noted a decline in their QoL at this stage, whereas surviving ILD patients without IPF started to improve at this stage. This enhances our knowledge of known prognostic indicators for different types of ILD. Improving QoL over the longer term in non-IPF ILD is an encouraging finding. The non-IPF ILD subgroup represents a clinically heterogeneous group. The observation of improvement in QoL in surviving patients is likely due to effective treatment options for certain NSIP subtypes.

Notably, surviving patients with IPF did not note a decline in their QoL until 48 months post diagnosis. This may reflect a milder disease phenotype in the surviving patients, or signal an abrupt symptomatic decline in patients with IPF in the couple of years before they die. A large registry study of 277 IPF patients observed SGRQ-measured QoL was static for 12 months and then declined in patients a mean of 3 years post-IPF diagnosis.[29] In particular, QoL has been shown to be very low in the 2 years prior to death in IPF patients.[30] Our study supports this existing work and reinforces the need for palliative care input alongside other treatments for IPF patients at this stage of disease.

## Limitations and future work

This study benefits from recruiting a large number of patients who have undergone PET-CT imaging and have been followed up for 4 years. Nonetheless, there are some limitations. After baseline QoL assessment, QoL data were only available for approximately 60% of surviving patients at each subsequent time point, which could introduce non-response bias. Furthermore, data were not collected on specific non-IPF ILD subtypes and as such we cannot draw conclusions on specific ILDs and the effect of $^{18}$F-FDG PET/CT as a prognostic marker. Finally, given the exploratory nature of this study, a formal sample size calculation was not used and as such the study could be underpowered to detect a clinically meaningful effect.

Future work could focus on PET/CT measures and QoL in specific ILD subgroups, particularly those with an inflammatory phenotype, and could explore the role of PET-CT in treatment response prediction.

## CONCLUSIONS

$^{18}$F-FDG PET/CT measure SUVmax demonstrated weak-to-moderate correlation with QoL at baseline, but baseline SUVmax and TBR were not associated with change in QoL in patients with IPF and non-IPF ILD. At present, there is insufficient evidence to support the role of $^{18}$F-FDG-PET/CT in QoL prediction in ILD. Further prospective studies of $^{18}$F-FDG-PET/CT involving treatment data, QoL and other imaging parameters may be beneficial to identify non-invasive disease markers.

At 4 years post-IPF diagnosis, surviving patients reported declining QoL, whereas those with non-IPF ILD reported an improvement in their QoL, potentially reflecting the beneficial effect of treatment.

**Author affiliations**
[1]University of Cambridge, Cambridge, UK
[2]Department of Respiratory Medicine, Cambridge University Hospitals, Cambridge, UK
[3]University College London Institute of Nuclear Medicine, London, UK
[4]ILD Centre, University College London Hospital, London, UK
[5]Department of Respiratory Medicine, University College London, London, UK
[6]Department of Respiratory Medicine, Lister Hospital, Stevenage, UK

**Contributors** LHJ contributed to the data analysis (statistical), data visualisation and drafted the original manuscript. BG contributed to data curation, data analysis (imaging and statistical), investigation, review and editing. LH contributed to data curation, investigation, visualisation, review and editing. IN contributed to review and editing. NM contributed to review and editing. AG contributed to conceptualisation, funding acquisition, methodology, visualisation, review and editing. JCP contributed to conceptualisation, funding acquisition, methodology, data curation, review and editing. TW contributed to conceptualisation, methodology, funding acquisition, project administration, supervision, review, editing and is the guarantor of the study. LHJ is the submitting and corresponding author.

**Funding** This study was undertaken at UCLH/UCL who received a proportion of funding from the Department of Health's NIHR Biomedical Research Centre's funding scheme.

**Competing interests** None declared.

**Patient and public involvement** Patients and/or the public were not involved in the design, or conduct, or reporting, or dissemination plans of this research.

**Patient consent for publication** Not applicable.

**Ethics approval** This study involves human participants and was approved by the Joint UCL/UCLH Committees on the Ethics of Human Research (Committee A)REC reference number 06/Q0505/22. Participants gave informed consent to participate in the study before taking part.

**Provenance and peer review** Not commissioned; externally peer reviewed.

**Data availability statement** Data are available on reasonable request. Data from this study are available on reasonable request. This comprises deidentified patient demographic and quality of life data and calculated PET parameters. Images from which PET parameters were calculated may also be made available on reasonable request.

**ORCID iD**
Louise Helen Jordon http://orcid.org/0000-0002-0066-4750

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
