## [Reviewer comments · BMJ Open]

ARTICLE DETAILS

TITLE (PROVISIONAL)	Can FDG-PET/CT imaging be used to predict decline in quality of life in Interstitial Lung Disease? A prospective study of the relationship between FDG uptake and quality of life in a UK outpatient setting.
AUTHORS	Jordon, Louise Helen; Ganeshan, Balaji; Nadeem, Iftikhar; Hoy, Luke; Mahdi, Noor; Porter, Joanna; Groves, Ashley; Win, Thida

VERSION 1 – REVIEW

REVIEWER	Choi , Joon Young Sungkyunkwan University School of Medicine
REVIEW RETURNED	23-Nov-2023

GENERAL COMMENTS	This study dealt with the associations between FDG PET/CT parameters and change in QoL in interstitial lung disease. Although this is an interesting topic, there were several critical issues in the Introduction and results for the consideration of publication. Title 1) Although the title dealt with prognosis, the primary outcome of this study was QoL. More appropriate title is necessary. Introduction 2) It is not clear why the authors investigated the association between baseline PET parameters and QoL. In interstitial lung disease, QoL depends on many factors including age, PFT, disease severity of interstitial lung disease and other diseases. Results 3) Other factors affecting on QoL should be included in the analysis. 4) The low returning rate of QoL questionnaire may be serious bias in this study. 5) Low correlation coefficients between PET parameters and change in QoL suggest negative results, and clinical value of this study is low.
---

REVIEWER	Justet, A Assistance Publique - Hôpitaux de Paris
REVIEW RETURNED	13-Dec-2023

GENERAL COMMENTS	Thank you for allowing me to review this manuscript.
--

	In this work, the authors aimed to investigate the relationship between 18FDG uptake and the quality of life in patients with fibrotic ILD, encompassing IPF and NSIP. The study holds several strengths worth highlighting, such as its prospective design and the inclusion of many patients. However, the paper is marred by several notable limitations: Limited Results and Further Analysis: The results presented are constrained, warranting additional analyses to fortify the manuscript's conclusions. Rationalization of PET Scan Use: Given the cost associated with PET scans, a more robust justification regarding the study's rationale is required. It's essential to discuss the relevance of PET CT scans in patients with fibrotic/progressive ILD. Lack of Treatment Information: The absence of treatment data significantly limits the study. Understanding its impact on lung 18FDG uptake is crucial, and it's unclear why this information couldn't be collected. Here are my detailed comments: Rationale Clarification: The rationale behind this study needs better articulation and justification. Even if a positive correlation between QOL and 18FDG uptake was found, it's important to elucidate the clinical relevance of performing an 18FDG PET scan for clinicians or patients. Exploration of Correlations: Considering the study design, exploring correlations between lung function decline and various scores alongside lung uptake would be beneficial to enrich the manuscript, particularly considering the negative and limited results. Marker Selection and Justification: The authors focused solely on SUV max and TBR, neglecting other markers like SUV mean which have been previously utilized. This approach needs justification in the discussion. Given the current results, including an analysis involving SUV mean could significantly enhance the manuscript's depth.
--	---

VERSION 1 – AUTHOR RESPONSE

Reviewer One: Although the title dealt with prognosis, the primary outcome of this study was QoL. More appropriate title is necessary.

We have amended the title such that it reflects the study outcomes more completely. The title now reads:

Can FDG-PET/CT imaging be used to predict decline in quality of life in Interstitial Lung Disease? A prospective study of the relationship between FDG uptake and quality of life.

Reviewer One: It is not clear why the authors investigated the association between baseline PET parameters and QoL. In interstitial lung disease, QoL depends on many factors including age, PFT, disease severity of interstitial lung disease and other diseases.

and

Reviewer Two: Rationalization of PET Scan Use: Given the cost associated with PET scans, a more robust justification regarding the study's rationale is required. It's essential to discuss the relevance of PET CT scans in patients with fibrotic/progressive ILD. Rationale Clarification: The rationale behind this study needs better articulation and justification. Even if a positive correlation between QOL and 18FDG uptake was found, it's important to elucidate the clinical relevance of performing an 18FDG PET scan for clinicians or patients.

Why is FDG-PET relevant in ILD?

There is an urgent need for better biomarkers in ILD – both for patient management and for research purposes. We have previously shown that FDG uptake on PET/CT is a prognostic biomarker in IPF (Win et al, 2018), where patients with a TBR below a threshold of 4.9 had a 3-year survival of >65% compared to a 3 year survival of <35% for patients with a TBR above the threshold. We also showed that utilising TBR helped to refine the GAP stage to predict outcomes.

To understand PET-CT as a biomarker further, we examined the correlation between FDG uptake and microvessel density on lung biopsy in fibrotic ILD, given that ILD is characterised by aberrant wound healing (Selman et al, 2003). We observed a significant positive correlation between the neoangiogenesis marker CD105, and TBR, both of which were shown to predict mortality in fibrotic ILD (Porter et al, 2022).

Most recently, we explored the relationship between FDG uptake and prognosis in NSIP, and found SUVmin (denoting high pulmonary background signal) to negatively correlate with mortality in 96 patients, and helped to refine patient outcome predictions when combined with the GAP score (Torlot et al, 2023).

Other groups have also investigated the utility of FDG uptake in ILD subtype differentiation (Nusair et al, 2007) and in determining disease activity in ILDs associated with rheumatological conditions (Peelen et al, 2019). Combined, this body of work demonstrates why FDG uptake is relevant in ILD studies.

Why explore QoL outcomes?

Traditionally, end points in clinical trials for ILD have focussed on pulmonary function tests (which correlate weakly with QoL), and mortality (which correlates with frailty, but not necessarily with QoL (Guler et al, 2020)). When asked, patients have reported outcomes which they value most to be symptom burden, daily functioning and QoL (Aronson et al, 2021). To that end, in 2021, the American Thoracic Society published a research statement encouraging the use of patient centred outcomes such as QoL to be focussed upon in clinical trials (Aronson et al, 2021).

As such, we wished to explore if FDG uptake, which has been shown to be a useful biomarker in ILD prognostication, would also be of benefit in predicting changes in QoL, a patient centered outcome which patients value most. Being able to predict changes in QoL would be of great benefit for patient management and holistic care provision.

We have amended the manuscript as below:

At diagnosis and during follow-up, it is difficult to predict prognosis for patients with ILD. When asked, patients have reported outcomes which they value most to be symptom burden, daily functioning and QoL [7]. Decline in Forced Vital Capacity (FVC) is most frequently used to monitor disease

progression [8], however baseline FVC is not predictive of change in QoL [5]. Routinely used biomarkers are lacking [9], and so tools to accurately predict QoL deterioration are needed.

Positron Emission Tomography (PET) combined with Computed Tomography (CT) offers a minimally invasive investigation to locate and quantify cellular metabolism. The most frequently used PET tracer is the glucose analogue, fluorodeoxyglucose (FDG). When labelled with the radioisotope 18-Fluorine (18F), it can be used to visualise areas of regional glucose uptake and indicate metabolic activity [10-12].

Tissue metabolic activity on PET-CT is quantified through Standardised Uptake Values (SUVs) which measure the tracer's radioactivity concentration relative to the dose given and the patient's weight [13]. The SUV is most commonly reported as the maximum SUV within the region of interest (SUVmax) [14]. The tissue-to-background ratio (TBR) is a measure of variation of FDG uptake within the lung, where a value close to one indicates relatively uniform uptake [15].

In IPF lung tissue, increased glycolysis has been observed in fibroblasts [16], with increased FDG uptake seen in areas of honeycombing [17, 18]. We have previously shown that FDG uptake on PET/CT is a prognostic biomarker in IPF [19], and that TBR is predictive of 3 year mortality. In patients with fibrotic ILD, we have observed that TBR correlates with neoangiogenesis markers [20], and in patients with NSIP we observed that a high pulmonary background signal (SUVmin) negatively correlated with mortality [21]. Others have investigated the utility of FDG uptake in ILD subtype differentiation [22] and in determining disease activity in ILDs associated with rheumatological conditions [23].

Reviewer One: Other factors affecting on QoL should be included in the analysis.

and

Reviewer Two: Exploration of Correlations: Considering the study design, exploring correlations between lung function decline and various scores alongside lung uptake would be beneficial to enrich the manuscript, particularly considering the negative and limited results.

We have previously shown baseline FDG uptake to be prognostic of mortality in IPF, and wished to specifically explore with whether QoL change could be predicted by baseline FDG uptake in ILD. The aim of this study was not to develop a new prognostic tool or model, with FDG uptake as an element of this, but to evaluate whether FDG uptake alone could be prognostic of change in QoL. This is of particular interest in QoL assessment, where we know that variables such as CT appearance and FVC do not neatly correlated with symptomatology.

As such, we have not designed a model involving multiple variables predicted to affect QoL as we wished to evaluate whether FDG uptake could be used alone in a predictive manner. Nonetheless, we agree that exploring correlations between variables such as pulmonary function tests and age with FDG uptake are of interest to the readers and as such have amended the manuscript as below:

Relationship between FDG-PET, SGRQ and other parameters

Age at enrolment did not correlate with baseline SUVmax ($r_s = -0.103$, $p=0.158$) or baseline TBR ($r_s = -0.033$, $p=0.646$).

Baseline FVC and TLCO data were available for 185 and 163 patients respectively. Baseline PET parameters did not correlate with baseline PFT data, nor did baseline PET parameters correlate with

change in PFT data after 12, 24 or 48 months of follow up. Further details are included in the supplemental materials (Table S2).

Table S2: Correlation between change in PFTs from baseline and baseline FDG-PET parameters
Correlation between change in FVC from baseline and SUVmax (number of participants)

	Correlation Coefficient (rs)	P value
Baseline (185)	-0.176	0.017
12 months (119)	0.010	0.912
24 months (73)	0.144	0.225
48 months (15)	0.220	0.427

Correlation between change in TLCO from baseline and SUVmax (number of participants)

	Correlation Coefficient (rs)	P value
Baseline (163)	-0.158	0.043
12 months (90)	0.072	0.503
24 months (52)	0.030	0.831
48 months (10)	0.185	0.604

Correlation between change in FVC from baseline and TBR (number of participants)

	Correlation Coefficient (rs)	P value
Baseline (185)	-0.003	0.967
12 months (119)	-0.101	0.273
24 months (73)	0.027	0.818
48 months (15)	-0.143	0.611

Correlation between change in TLCO from baseline and TBR (number of participants)

	Correlation Coefficient (rs)	P value
Baseline (163)	-0.036	0.648
12 months (90)	-0.052	0.625
24 months (52)	-0.046	0.748
48 months (10)	-0.098	0.789

There was weak negative correlation between baseline FVC and baseline TLCO and total baseline SGRQ score (FVC: $rs=-0.285$, $p<0.001$, TLCO: $rs=-0.214$, $p=0.006$). Change in PFT measures did not correlate with change in total SGRQ score over the 48 month follow up period.

Reviewer One: The low returning rate of QoL questionnaire may be serious bias in this study.

At each subsequent time point from baseline, the QoL questionnaire was returned by approximately 60% of surviving patients. As such, we agree that there is potential for nonresponse bias to impact the results. Nonetheless, a survey response rate in this region, is in line with that acceptable for publication in many journals (e.g. JAMA: <https://jamanetwork.com/journals/jama/pages/instructions-for-authors>), and see Meterko et al, 2015.

To make it clearer that the low returning rate of the questionnaires was in part due to patients dying during the follow up period, we have amended the manuscript as below:

QoL data was available for all 193 patients at the time of recruitment (T0), 99 patients at T12 (57.2% surviving patients), 91 patients at T24 (60.3% surviving patients) and 45 patients at T48 (61.6% surviving patients).

We have also added the following to the limitations section:

This study benefits from recruiting a large number of patients who have undergone PET-CT imaging and been followed up for 4 years. Nonetheless, there are some limitations. After baseline QoL assessment, QoL data was only available for approximately 60% of surviving patients at each subsequent time point, which could introduce nonresponse bias.

Reviewer One: Low correlation coefficients between PET parameters and change in QoL suggest negative results, and clinical value of this study is low.

Work continues to find a biomarker/variable which can predict change in QoL in patients with ILD over time. QoL outcomes are those most cared about by patients. This study demonstrates that FDG uptake, whilst a useful prognostic biomarker for other ILD outcomes such as mortality, cannot be used to predict QoL decline. We believe that it is important to communicate negative findings such as these to the wider scientific and clinical communities to aid in the finding of such a biomarker/variable in the future.

Reviewer Two: Lack of Treatment Information: The absence of treatment data significantly limits the study. Understanding its impact on lung 18FDG uptake is crucial, and it's unclear why this information couldn't be collected.

We agree that the impact of treatment at baseline on FDG uptake at baseline is of interest. Whilst treatment data was not specifically collected for purposes of this study, complete treatment data was available for 127 patients (127/193; 66%). An analysis was performed on this subgroup to determine the impact of ILD treatments on FDG uptake.

The manuscript has been amended as below:

ILD treatment data was available for 127 participants at baseline (127/193; 66%). Of these, 84 participants were on no treatment, 35 were on steroid treatment, 9 were on antifibrotic treatments and 10 were on immunosuppressant treatment. 13 patients were on more than one treatment at baseline, 11 of which comprised treatments in different categories. In this subgroup of patients where treatment data was available, type of treatment (or no treatment) at baseline did not affect baseline SUVmax ($H(4)=0.88$, $p=0.83$) or SUVmin ($H(4)=4.41$, $p=0.22$). There was a significant difference in median TBR between the treatment groups (Median TBR by group: No treatment = 6.0, Steroid = 5.0, Antifibrotic = 5.3, Immunosuppressant = 4.0, $H(4)=8.54$, $p=0.036$).

Given the small number of patients in the immunosuppressant and antifibrotic treatment groups, the impact of this must be considered when interpreting the TBR results.

Reviewer Two: Marker Selection and Justification: The authors focused solely on SUV max and TBR, neglecting other markers like SUV mean which have been previously utilized. This approach needs justification in the discussion. Given the current results, including an analysis involving SUV mean could significantly enhance the manuscript's depth.

From our previous work in the area, as referred to in previous responses, we have found the markers TBR, SUVmin and SUVmax to be superior to SUVmean in ILD prognostication. As such we wished to evaluate whether these markers were predictive of QoL change. Furthermore, Consensus recommendations on PET/CT use in lung disease suggest using SUVmax as the quantitative variable (Chen et al, 2020), and as such this parameter is most clinically relevant, and not confined to the research setting.

We did collect SUVmin data, and have added it to our manuscript in the results section. An example section of the amended results is below:

Relationship between FDG-PET and SGRQ

Example PET-CT images with their corresponding SGRQ scores are shown in Figure 1A-D.

At baseline, there was weak positive correlation between SUVmax and total SGRQ score ($r_s=0.229$, $p=0.002$) and the SGRQ Symptom, Activity, and Impact subscales ($r_s=0.211$, 0.178 and 0.209 respectively, $p<0.003$ for all). There was also weak positive correlation between SUVmin and baseline total SGRQ score ($r_s=0.233$, $p=0.001$) and the SGRQ Symptom, Activity, and Impact subscales ($r_s=0.161$, 0.153 and 0.236 respectively, $p<0.05$ for all). There was no significant relationship between TBR and total ($r_s=-0.015$, $p=0.841$) and subscale SGRQ scores.

When change in QoL over time was assessed, there was no significant correlation between change in total SGRQ score and baseline SUVmax at 12 months ($r_s = -0.084$, $p=0.403$), 24 months ($r_s=0.184$, $p=0.084$) and 48 months ($r_s=0.084$, $p=0.56$). Additionally, there was no relationship between change in SGRQ subscale scores at each time point and SUVmax. There was no significant correlation between change in total SGRQ score at 12 months ($r_s=-0.017$, $p=0.863$), 24 months ($r_s=0.122$, $p=0.253$ and 48 months ($r_s=0.140$, $p=0.332$) and baseline SUVmin. We observed no significant correlation between change in total SGRQ score and baseline TBR at T12 ($r_s= -0.065$, $p=0.519$), T24 ($r_s= -0.014$, $p=0.893$) and T48 ($r_s= -0.113$, $p=0.435$), and no relationship between change in SGRQ subscale scores at each time point and TBR.

Editor comments

*Please ensure that all abbreviations are defined on first mention, including those in the Abstract.

PET has now been defined in the abstract.

*Along with your revised manuscript, please include a copy of the checklist that would be most appropriate for your study, indicating the page/line numbers of your manuscript where the relevant information can be found (E.g. STROBE checklist: <https://strobe-statement.org/index.php?id=strobe-home>).

A STROBE checklist has been completed and uploaded.

We thank you for your consideration of our revised manuscript

Kind regards

Dr Louise Jordon
On behalf of the authors

References:

Aronson K, Danoff S, Russell A, Ryerson C, Suzuki A, Wijsenbeck M et al. Patient centered outcomes research in Interstitial Lung Disease: An official American Thoracic Society research statement. *American Journal Respiratory and Critical Care Medicine*. 2021; 204(2): 3-23. DOI 10.1164/rccm.202105-1193ST.

Chen D, Ballout S, Chen L, Cheriyan J, Choudray G, Denis-Baclear A, Emnd E, Erlandsson K et al. Consensus recommendations on the use of 18F-FDG PET/CT in lung disease. *Journal of Nuclear Medicine*. 2020; 61(12): 1701-1707. DOI: <https://doi.org/10.2967/jnumed.120.244780>

Guler S, Kwan J, Leung J, Khalil N, Wilcox P and Ryerson C. Functional ageing in fibrotic interstitial lung disease: the impact of frailty on adverse health outcomes. *European Respiratory Journal*. 2020; 55: 1900647; DOI: 10.1183/13993003.00647-2019

Meterko M, Restuccia J, Stolzmann K, Mohr D, Brennan C, Glasgow J and Kaboli P. Response rates, nonresponse bias and data quality: results from a national survey of senior healthcare leaders. *Public Opinion Quarterly*. 2015; 79(1): 130-144.

Nusair S, Rubinstein R, Freedman NM, Amir G, Bogot NR, Izhar U. Positron emission tomography in interstitial lung disease. *Respirology*. 2007;12(6):843–847. doi: 10.1111/j.1440-1843.2007.01143.x.

Peelen D, Zwezijnen B, Nossent E, Meijboom L, Hoekstra O et al. The qualitative assessment of interstitial lung disease with positron emission tomography scanning in systemic sclerosis patients. *Rheumatology*. 2020; 59(6): 1407-15.

Porter J, Ganeshan B, Win T, Fraioli F, Khan S, Rodrigues-Justo M et al. 18F-FDG PET signal correlates with markers of neo-angiogenesis in patients with fibrotic interstitial lung disease who underwent lung biopsy: implication for the use of PET/CT in diffuse lung diseases. *Research Square* [PrePrint] <https://doi.org/10.21203/rs.3.rs-1673011/v1>

Selman M, Pardo A. The epithelial/fibroblastic pathway in the pathogenesis of idiopathic pulmonary fibrosis. *American Journal of Respiratory Cell and Molecular Biology*. 2003; 29(3): S93-97.

Torlot H, Win T, Ganeshan B, Sreaton N, Maher T, Fraiolo F, Endozo R, Shortman R et al. Pulmonary 18F-FDG uptake refines current risk stratification in non-specific interstitial pneumonitis (NSIP) and predicts response to treatment. *European Respiratory Journal*. 2023 62: PA2920. DOI: 10.1183/13993003.congress-2023.PA2920